# Genetic and Phenotypic Characterization of Soybean Landraces Collected from the Zhejiang Province in China

**DOI:** 10.3390/plants13030353

**Published:** 2024-01-25

**Authors:** Xiaomin Yu, Xujun Fu, Qinghua Yang, Hangxia Jin, Longming Zhu, Fengjie Yuan

**Affiliations:** 1Institute of Crop and Nuclear Technology Utilization, Zhejiang Academy of Agricultural Sciences, Hangzhou 310021, China; fuxj@zaas.ac.cn (X.F.); yangqf@zaas.ac.cn (Q.Y.); jinhx@zaas.ac.cn (H.J.); zlmsllzly@163.com (L.Z.); 2Xianghu Laboratory, Hangzhou 311231, China; 3Key Laboratory of Information Traceability for Agricultural Products, Ministry of Agriculture and Rural Affairs of China, Zhejiang Academy of Agricultural Sciences, Hangzhou 310021, China; 4Key Laboratory of Digital Upland Crops of Zhejiang Province, Zhejiang Academy of Agricultural Sciences, Hangzhou 310021, China

**Keywords:** soybean, landrace, germplasm resource, genetic diversity, phenotypic variation

## Abstract

The soybean is an important feed, industrial raw material, and food crop in the world due to its rich components. There is a long history of soybean cultivation with different types and rich resources in the Zhejiang province of China. It is important to understand genetic diversity as well as phenotypic variation for soybean breeding. The objective of this study was to analyze both genetic and phenotypic characteristics of the 78 soybean landraces collected, and to explore a potential advantage of germplasm resources for further application. These 78 autumn-type soybean landraces have been propagated, identified, and evaluated in both 2021 and 2022. There were agronomic, quality, and genetic variations according to the comprehensive analyses. There was a good consistency between seed size and seed coat color. There were significant differences of seed protein, fat, and sugar contents based upon the seed coat color. These soybean landraces were genotyped using 42 simple sequence repeat markers and then clustered into two groups. The two groups had a consistency with the seed coat color. This study gave us a combined understanding of both the phenotypic variation and the genetic diversity of the soybean landraces. Therefore, the reasonable crossing between different soybean types is highly recommended.

## 1. Introduction

The soybean (*Glycine max*) is a legume species that is native to China and known all over the world due to its history of more than 5000 years [1,2,3]. Because of its protein, oil, isoflavones, unsaturated fatty acids, vitamins, and other nutrients, soybean has become an important crop as a food, oil, and forage [2,3,4]. There have been more than 43,000 soybean germplasm resources preserved in the National Crop Germplasm Resource Bank of China (Beijing), including 8518 wild soybean (*Glycine soja*) germplasm resources and 3265 imported germplasm resources from abroad [2,5,6]. The evaluation of germplasm resources is a frequent research topic. It is a basic requirement for the resource research to meet both breeding and production needs, according to the development and changing nature of agricultural production [2,7]. Therefore, the main agronomic seed traits (such as seed coat color, seed shape, cotyledon color, and 100-seed weight) of all these soybean resources in the catalog have been investigated and recorded in order to reflect the variety and characteristics. The quality analyses of soybean grain have also been conducted for these resources, including the contents of protein, oil, isoflavones, lipoxygenases, and amino acids. Furthermore, the resistance to diseases, insects, and other stressors has also been studied among some of these soybean resources [2,5,6,8].

The main components of these germplasm resources are soybean landraces (including improved cultivars and breeding lines) collected from each province in China. Although these soybean landraces have a strong adaptability, most of their comprehensive traits are not as good as the commercial cultivars and thus are not directly suitable for agricultural production [2]. Therefore, it is necessary to reform germplasm innovation because the landraces have rarely been directly used as parental materials in breeding. The germplasm innovation would not only enrich the number of germplasm resources, but also improve and develop the quality of germplasm resources [7,8,9,10]. Therefore, the germplasm innovation is an indispensable aspect of the resource research. While many breeders are selecting new varieties, they are also innovating their germplasm by retaining certain strains with outstanding characteristics and utilizing them in breeding. For example, breeders often used yellow-seeded soybeans with high combining ability and good comprehensive traits as the parents for breeding new varieties in the past, resulting in a narrower genetic basis for soybean varieties. Therefore, it is necessary to further explore the high-yield genes of local varieties and closely-related wild species, so as to introduce them into the existing high-yield varieties. It is also necessary to utilize the germplasm of green-seeded soybeans, black-seeded soybeans, and brown-seeded soybeans in order to continuously expand the genetic foundation of different varieties [2,7].

The Zhejiang province is located on the southeast coast of China and has a variety of landforms, climates, and crops. Grain and oil crops in dryland play an important role in agricultural production in this province, among which soybeans have one of the largest planting areas [11]. Due to the different harvesting times and applications, it could be divided into vegetable type (as fresh vegetable for consumption) and grain type (as raw materials for processing) [4,12,13]. There is a long history of soybean cultivation with different types and rich resources, including spring, summer, and autumn types, based upon the sowing season. Spring soybeans are typically sown from late March to early April and harvested from late June to early July. Summer soybeans are sown from late June to early July and harvested in late October. Autumn soybeans are sown from late July to early August and harvested in early November [3,11]. Zhejiang is one of the provinces with a large number of the autumn soybean resources collected and preserved in China [2,3,11]. These landraces contain rich genetic resources, which are the foundation for the promotion and development of the characteristic soybean industry in this locality. Therefore, it is a current area of concern in terms of how to transform this potential advantage into a practical advantage [9,10,14,15].

A total of about 78 soybean landraces (autumn type) have recently been collected in Zhejiang. This study focuses on these 78 newly collected soybean landraces with different seed coat colors and seed sizes. They were propagated, identified, and evaluated between 2021 and 2022, and their agronomic, quality, and genetic variations were comprehensively analyzed. This could provide a scientific basis for the protection and utilization of soybean germplasm resources in the Zhejiang province and even the whole country.

## 2. Results

### 2.1. Genetic Variation

In this study, a total of 78 soybean landraces were genotyped using 42 simple sequence repeat (SSR) markers that were evenly distributed in the well-established soybean linkage groups [16,17]. All SSR markers provided unambiguous bands and gave a total of 509 alleles across all landraces, with an average of 12.1 alleles per locus (Table 1). There were 29 (69.0%) loci with the number of alleles being more than 10. The least polymorphic marker, satt309, and the most polymorphic marker, satt281, amplified four and twenty-three alleles, respectively. Among these alleles, 22.8% (116 of 509) were found to be unique to only one landrace.

The polymorphism information content (PIC) value is a reflection of allelic diversity and frequency among the landraces analyzed [9,10]. In this study, the PIC values for 42 SSR markers ranged from 0.211 to 0.930, with an average of 0.739 and the majority being between 0.650 and 0.850 (Table 1). The lowest PIC value was found in the marker satt571, and the highest PIC value was found in the marker satt281. Giving consideration to both the allele number and the PIC value, more than half of the markers in this study were noteworthy due to the simultaneously higher PIC value and the relatively higher levels of polymorphism.

### 2.2. Agronomic Characteristics

To identify the differences of seed characteristics among the seventy-eight soybean landraces, six seed traits were investigated, including seed coat color, hilum color, cotyledon color, seed coat luster, seed shape, and 100-seed weight. Among these landraces, 33, 26, and 19 landraces had yellow, black, and green seed coat colors, respectively. The cotyledon colors of the 59 landraces were yellow and the other 19 landraces showed green cotyledon colors. The 100-seed weights ranged from 9.5 to 50.5 g with a mean of 32.2 g (Figure 1). To further categorize these 78 landraces, the means of 100-seed weights for yellow, black, and green-seeded soybeans were 29.3, 32.9, and 36.0 g, respectively (Figure 2). The heaviest 100-seed weight was identified in Jiashan Madou (Z157), whereas the lightest 100-seed weight was detected in Kaihua Heidou (Z073).

Furthermore, the growth durations of the 78 soybean landraces ranged from 75 to 105 d, with a mean of 95.8 d (Figure 1). Among them, three landraces, including Chunan Qingdou (Z017), Tiantai Huangdou (Z057), and Ninghai Huangdou (Z009), exhibited an early maturity, with less than 80 d at the growth duration. And the growth durations of about 12 landraces were longer than 100 d. In addition, the plant heights of the total landraces ranged from 40 to 108 cm, with a mean of 68.6 cm (Figure 1). The lodging is closely associated with the plant height and is also an important factor influencing the soybean yield. However, these 78 soybean landraces were susceptible to lodging at different degrees, which potentially results in yield reduction and harvest losses.

### 2.3. Quality Traits

To gain insight into a quality profile of the soybean seeds, we analyzed protein, oil, and sugar contents in the seeds of these 78 soybean landraces. The contents of protein ranged from 34.94% to 45.29%, with a mean of 39.58%; oil contents ranged from 16.47% to 20.81%, with a mean of 18.64%; and sugar contents ranged from 1.01% to 14.05%, with a mean of 7.05% (Figure 3). The means of the protein contents for yellow, black, and green-seeded soybeans were 41.77%, 37.76%, and 38.26%, respectively. The means of the oil contents for yellow, black, and green-seeded soybeans were 17.95%, 19.56%, and 18.58%, respectively. The means of the sugar contents for yellow, black, and green-seeded soybeans were 9.76%, 3.68%, and 6.94%, respectively (Figure 2). One landrace, Qingyuan Huangdou (Z092), had the highest protein and lowest oil contents. The lowest protein content was found in Linan Heidou (Z115), and the highest oil content was detected in Wuyi Wudou (Z116). The highest sugar content was identified in Ruian Huangdou (Z025), whereas the lowest sugar content was detected in Lanxi Wudou (Z063).

Correlation analysis was further performed among the 100-seed weights and seed protein, oil, and sugar contents (Table 2). The correlations were statistically significant between 100-seed weight and protein contents (r = −0.391), 100-seed weight and oil contents (r = 0.419), protein and oil contents (r = −0.740), protein and sugar contents (r = 0.461), as well as oil and sugar contents (r = −0.449). However, the correlation was close to zero between 100-seed weight and sugar content. The correlations revealed in this study between the seed size and the protein/oil content provide new information for soybean breeding for improving both the seed size and quality traits.

### 2.4. Cluster Analysis

Based upon the genotyping data of 42 SSR markers, the principal component analysis (PCA) showed the relatively close relationships between the landraces, in which the majority were distributed into two sub-populations, although with several outliers (Figure 4). The unweighted pair group method using arithmetic mean cluster analysis further indicated that these soybean landraces could be classified into two groups when combined with the PCA result (Figure 5). These two groups showed a great consistency with the seed coat color. Group I consisted of 36 landraces, and 33 of them had the yellow seed coat. It is interesting to point out that the landraces from group II showed the black or green seed color but has various agronomic performances. Among them, there were seven landraces with the green seed color, often applied as the vegetable type in the locality. In addition, six soybean landraces were recognized as the outliers, including two lines with the green seed coat and four lines with the black seed coat. However, their agronomic performances that we observed were not particular when compared with the other landraces. Hence, more phenotypic identification is needed since these soybean landraces might have a different pedigree and come from other provinces.

## 3. Discussion

It is informative and effective to use SSR markers as a tool to analyze the genetic diversity in soybeans [7,14,16,17]. A total of 42 SSR markers were employed in genotyping; the average number of alleles per loci was 12.1 and the average PIC value was 0.739. These two parameters were much higher in this study than those in the previous studies [9,10,18,19]. We also found that 13 markers amplified less than 10 alleles in this study. These differences may declare that the polymorphism of SSR loci is relative and depends on the genetic populations analyzed.

The cultivation of soybeans has a long history in Zhejiang, and a rich and diverse variety of soybean germplasm resources have been formed through the natural and artificial selection during a long-term growing process [11,17,20]. However, the quantity of varieties planted has become increasingly limited with the application and promotion of current breeding lines [2,11,17]. Therefore, most landraces are in danger of extinction, and the reduction of germplasm resources is relatively obvious in Zhejiang. The breeding parents are often limited to a few varieties with good comprehensive traits, resulting in an extremely low utilization of the germplasm resources in new variety breeding.

The genotyping and phenotyping results showed that soybean germplasms were relatively diverse in Zhejiang [11,17,20]. The excessive usage of limited elite lines owned by each breeder may lead to a relatively narrowed genetic background and will be adverse to the soybean industry. The genetic basis of the breeding varieties was narrowed, making it difficult to achieve a breakthrough in both the yield and important quality traits [2,7]. This study gave us a visualized understanding of the genetic diversity and phenotypic variation of the soybean landraces assessed.

The reasonable crossing of germplasms is highly recommended with different groups or outliers, based on this result. For example, the autumn soybean landrace, Deqing Hedou with a black seed coat color, was used as the female parent and crossed with the Huanghuai summer soybean, Yanhuang No.1, with a yellow seed coat color, to produce a new variety, Zhechun No.2 [2,21]. This variety has good adaptability, disease resistance, and aluminum-ion resistance, and it has been promoted in red and yellow soil areas in the south, such as Zhejiang, Jiangxi, Fujian, and even Sichuan and Yunnan, indicating a great potential for germplasm resource utilization. Therefore, the utilization of landraces with a superior performance for breeding is highly encouraged.

In this study, there were a few soybean landraces with superior quality traits when compared with the commercial cultivars and breeding lines. One yellow-seeded landrace, Qingyuan Huangdou (Z092), had a protein content of more than 45%. Two black-seeded landraces, Wuyi Wudou (Z116) and Longyou Heidou (Z124), were often planted for bean curd production, having oil contents of about 21%. Two green-seeded landraces, Ninghai Qingdou (Z018) and Jiashan Maodou (Z157), were commonly identified as edamame with a good flavor and a large seed size (100-seed weight about 50 g). These soybean landraces present potential for utilization as breeding parents. However, it might take a long time to get rid of some of their agronomic traits (such as seed coat color and plant lodging) and would therefore be a challenge for any further direct application in soybean breeding. Therefore, these soybean landraces could be utilized as a material basis for breeding development.

## 4. Materials and Methods

### 4.1. Plant Materials

A group of the 78 soybean landraces (traditional cultivars) collected from Zhejiang province were recently selected to represent the autumn soybean germplasms in southern China [2,17,22]. These soybean landraces were grown in a randomized complete block design with three replicates in late July of 2021 and 2022, in the experimental field of Zhejiang Academy of Agricultural Sciences (Hangzhou, China). For each replicate, three-row plots, 5 m long and 0.5 m row-spaced, were planted at a rate of 10 plants per meter. According to the local planting habits, the leave samples were collected a month after seedling emergence. The seeds were harvested at a natural maturity and then air dried for storage for each year [3]. Some agronomic traits were recorded, including growth duration, plant height, and lodging (Appendix A). The levels of lodging were described as erect, light prostrate, significant prostrate, severe prostrate, or prostrate, based on the rate of plant lodging. The growth duration was calculated as the number of days from the day of seed emergence to seed maturity. The plant height was calculated as the height from the cotyledon node to the top of the plant (excluding the inflorescence), expressed in centimeters (cm) [23].

### 4.2. SSR Analysis

Genomic DNA was isolated from the young leaves of each landrace using a CTAB method [3,16]. The SSR analysis was performed according to the previous reports [24,25]. Based upon these previous reports [3,16,24,25], 42 SSR markers, which were evenly distributed on 20 soybean genetic linkage groups, were used in the genotyping. The primers were then synthesized (Sunny Biotech, Shanghai, China) and the polymerase chain reaction (PCR) reaction was performed (Eppendorf Corporation, Hamburg, Germany).

The PCR mixture consisted of 100 ng genomic DNA, 0.3 μM each primer, 10 μL 2X GoTaq Green Mix (Promega Corporation, Fitchburg, WI, USA), and was then brought up to 20 μL of total volume with ddH_2_O. To remit the disadvantage of various annealing temperatures, a touchdown process was applied for these different primer pairs. The thermal cycling procedure was set as follows: pre-denatured at 94 °C for 3 min; followed by 12 cycles of 94 °C for 40 s, 60 °C for 30 s, 72 °C for 40 s, and the annealing temperature was decreased by 1 °C for each cycle; followed by 25 cycles of 94 °C for 40 s, 48 °C for 30 s, 72 °C for 40 s; PCR products were finally extended at 72 °C for 7 min and then store at 4 °C for further analysis.

The PCR products were separated using 4.7% denaturing urea polyacrylamide gel electrophoresis (Bio-Rad Corporation, Hercules, CA, USA). The PAGE gels were stained in 0.1% silver nitrate solution for 10 min, washed for 30 s, and then visualized in a 2% NaOH solution with 0.5% formaldehyde until the DNA bands appeared clearly. A zero-one data matrix was created, according to the absence or presence of each band for each cultivar. The unweighted pair group method using arithmetic mean cluster analysis was then performed to construct a distance tree [16,17,18,19].

### 4.3. Seed Trait Analysis

According to the seeds harvested each year, the seed characteristics were recorded for each landrace, including the colors of the seed coat, hilum, and cotyledon, as well as the seed shape, size, and the seed coat luster (Appendix A). The seed coat color was described as yellow, green, black, brown, or di-color. The seed hilum color was described as yellow, black, or brown. The seed cotyledon color was described as yellow or green. The seed shape was described as spherical, reniform, ellipse, flat ellipse, long ellipse, or spherical flattened. The seed coat luster was described as glimmer or absent [23]. The 100-seed weight was also calculated in triplicate with random sampling and then averaged within two years [15].

### 4.4. Seed Quality Analysis

Seed samples of about 50 g from each soybean landrace were analyzed by near-infrared reflectance (NIR) spectroscopy, using an Infratec 1241 Grain Analyzer (Foss Corporation, Stockholm, Sweden). The total value of each landrace used for the data analysis was an average of ten sub-samples, measured by NIR. The contents of the protein, oil, and sugar were presented as grams per 100 g (%) of dry seed matter units (Appendix A). The statistical analyses, including PCA, correlation, and cluster analyses, were conducted using SPSS statistics 19.0 (IBM, Armonk, NY, USA) [3,7,15,16,22,26].

## 5. Conclusions

The Zhejiang province has a long history of soybean cultivation with the different types of rich resources in China. In this study, a total of 78 autumn-type soybean landraces have been collected and evaluated across two years. The comprehensive analyses showed the agronomy, quality, and genetic variations. The seed size exhibited a distinction between the landraces with different seed coat colors. In addition, there were significant differences between seed protein, fat, and sugar contents based upon the seed coat color. Meanwhile, these soybean landraces were genotyped using 42 SSR markers and then clustered into two groups. The two groups were consistent when regarding the seed coat color. This study gave us a clear understanding of both the genetic diversity and phenotypic variation of soybean landraces in Zhejiang. Therefore, the application of a reasonable crossing of soybean landraces between different types is highly recommended.

## Figures and Tables

**Figure 1 plants-13-00353-f001:**
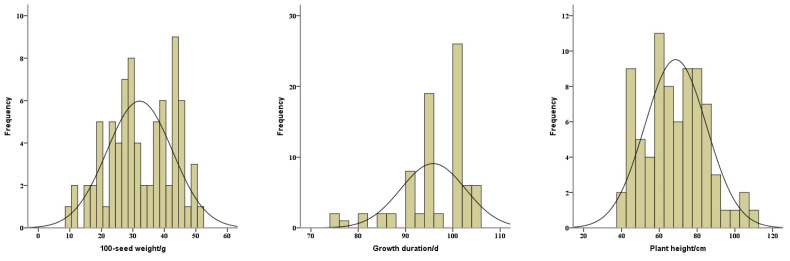
Distributions of 100-seed weight (**left**), growth duration (**middle**), and plant height (**right**) of the total soybean landraces.

**Figure 2 plants-13-00353-f002:**
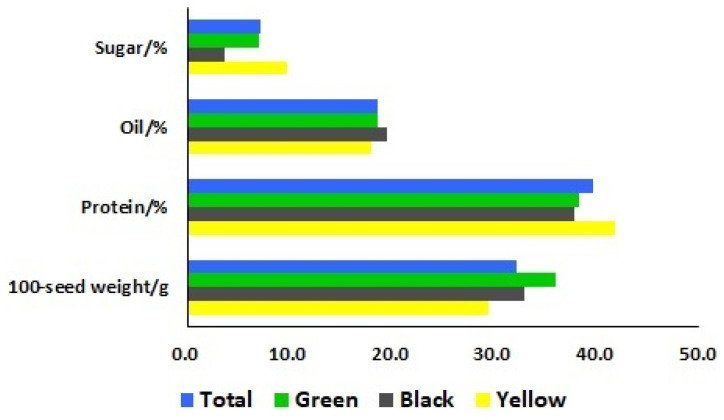
Means of seed sugar, oil, and protein contents and the 100-seed weight of the total soybean landraces, as well as three different seed coat colors (green, black, and yellow).

**Figure 3 plants-13-00353-f003:**
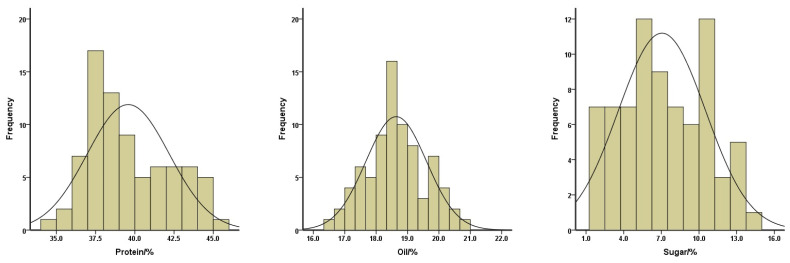
Distributions of the protein (**left**), oil (**middle**), and sugar (**right**) contents of the total soybean landraces.

**Figure 4 plants-13-00353-f004:**
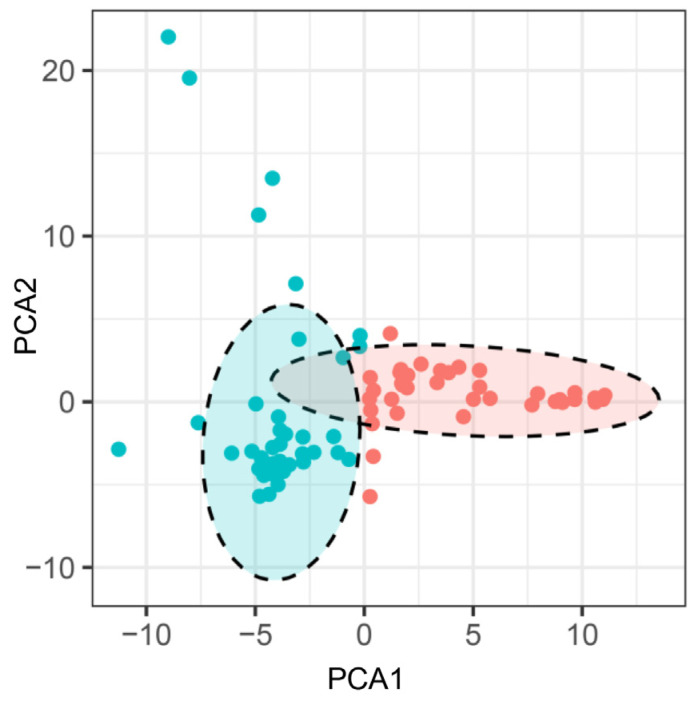
Principal component analysis of the soybean landraces based on the SSR markers. **Note:** Each dot represents an individual landrace. Dots with the same color belong to one sub-population.

**Figure 5 plants-13-00353-f005:**
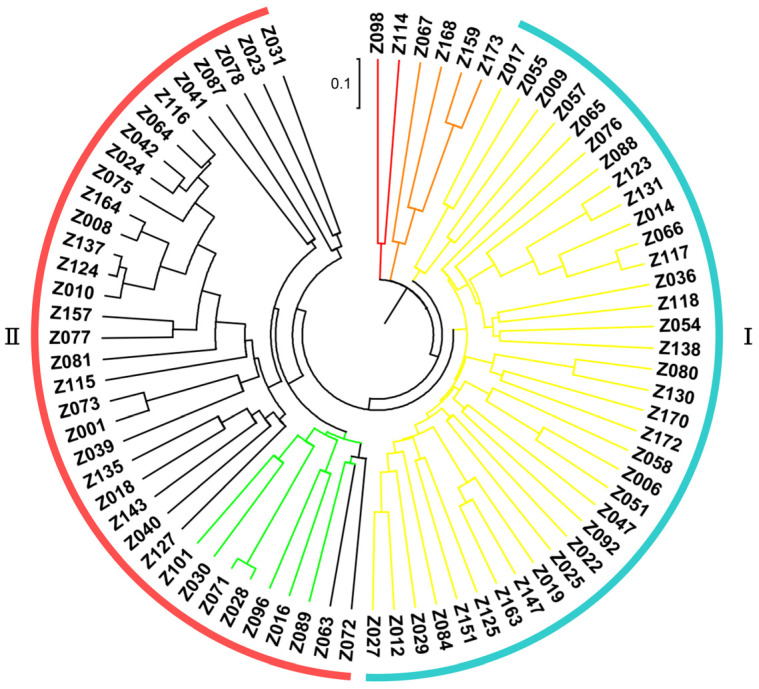
Clustering of the soybean landraces based on the SSR markers.

**Table 1 plants-13-00353-t001:** SSR locus, linkage group with position, chromosome number, no. of alleles found, gene diversity, heterozygosity, and polymorphism information content (PIC) of all markers.

Primer Name	Linkage Group	Position (cM)	Chromosome Number	No. of Alleles	Gene Diversity	Heterozygosity	PIC
satt300	A1	30.93	5	9	0.437	0.092	0.426
satt236	A1	80.81	5	12	0.848	0.077	0.832
satt390	A2	7.43	8	8	0.723	0.065	0.687
satt409	A2	145.57	8	14	0.805	0.321	0.790
satt197	B1	49.07	11	19	0.851	0.246	0.837
satt168	B2	46.87	14	8	0.740	0.066	0.704
satt556	B2	63.25	14	11	0.660	0.053	0.631
satt565	C1	5.74	4	13	0.686	0.053	0.642
satt194	C1	13.37	4	9	0.656	0.189	0.595
satt281	C2	38.90	6	23	0.934	0.325	0.930
satt286	C2	92.48	6	10	0.841	0.128	0.822
satt307	C2	109.96	6	8	0.781	0.120	0.749
satt184	D1a	16.13	1	13	0.718	0.143	0.698
satt267	D1a	40.55	1	10	0.793	0.068	0.770
satt157	D1b	46.23	2	20	0.918	0.205	0.913
satt005	D1b	83.41	2	20	0.878	0.299	0.870
satt002	D2	42.69	17	13	0.822	0.130	0.804
satt230	E	71.30	15	5	0.590	0.042	0.507
satt586	F	33.70	13	11	0.800	0.064	0.778
satt146	F	37.14	13	16	0.877	0.092	0.867
satt334	F	51.20	13	13	0.712	0.154	0.687
satt309	G	10.10	18	4	0.581	0.014	0.495
satt352	G	50.52	18	8	0.714	0.053	0.686
satt442	H	43.39	12	16	0.897	0.256	0.888
satt279	H	63.49	12	10	0.657	0.039	0.637
satt571	I	14.97	20	5	0.218	0.014	0.211
satt239	I	29.61	20	6	0.734	0.039	0.688
sct_189	I	109.27	20	11	0.813	0.055	0.791
satt414	J	37.04	16	16	0.832	0.090	0.821
satt596	J	39.63	16	19	0.890	0.055	0.882
satt215	J	47.36	16	6	0.650	0.026	0.586
satt431	J	82.03	16	10	0.777	0.107	0.753
satt242	K	14.73	9	9	0.717	0.301	0.679
satt588	K	93.44	9	17	0.878	0.068	0.866
satt373	L	93.95	19	21	0.884	0.167	0.875
satt590	M	7.75	7	16	0.882	0.133	0.872
satt346	M	106.12	7	9	0.775	0.051	0.745
satt308	M	122.55	7	11	0.820	0.107	0.800
satt530	N	32.84	3	13	0.872	0.090	0.859
satt339	N	60.17	3	17	0.897	0.231	0.888
satt022	N	84.45	3	10	0.818	0.053	0.799
satt243	O	107.31	10	10	0.716	0.095	0.668

**Table 2 plants-13-00353-t002:** Correlations of seed protein, oil and sugar contents, and 100-seed weight.

	100-Seed Weight	Protein	Oil	Sugar
100-seed weight	1.000	−0.391 **	0.419 **	0.068
Protein		1.000	−0.740 **	0.461 **
Oil			1.000	−0.449 **
Sugar				1.000

** Significant at the 0.01 probability level.

## Data Availability

The data generated in this study are included in this published article and its Appendix A.

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
