# Peer review of "Genetic and Phenotypic Characterization of Soybean Landraces Collected from the Zhejiang Province in China"

_plants, 2024, doi:10.3390/plants13030353_

Round 1
Reviewer 1 Report
Comments and Suggestions for Authors
1. To demonstrate the usefulness of this study, I wonder why the authors did not include any of the current commercial cultivars into the field evaluation. Without knowing the 100-seed weight and seed quality traits of the commercial cultivars we will not be able to desmontrate if these 78 landraces would be useful for crossing and cultivar improvement. Please explain why you did not include benchmark cultivars in your study.
2. The authors stated that you collected agronomic data, but mostly seed quality traits recorded. Important agronomic traits such as maturity, height, flowering time, lodging were not included. Either you leave “agronomic” words out so your paper only focuses on seed quality traits, or you have to include those agronomic traits if you happen to collect those traits as well.
3. There was many data collected, but the authors were not be able to present data in a more informative means rather than just table, min, max and average. I suggested to present data in histograms (again, it would be nice if you have some benchmark cultivars as controls to show lines that surpass your controls in phenotypic values that can be used by breeders, but I suspect you do not have them here), correlation matrices (protein, oil, sugar content and 100-seed weight). Bargraphs for 100-seed weight, protein, oil and sugar contents for different seed coat colour groups as it seems soybeans with black seed coat tend to be heavier than those of yellow and green ones.
Line 100-127: does the max values that you found much higher than those of the commercial varieties being sown currently?
Line 188: some locus has 10 alleles or more, then the codes for alleles just 0 and 1? I thought it can be done from 1-10 for examples, that may help improve the clustering?
Comments on the Quality of English Language
English language was mostly fine, but can be improved and polished one more time to add in the, a or plural at some places
Reviewer 2 Report
Comments and Suggestions for Authors
Comments for the manuscript entitled “Genetic and Phenotypic Characterization of Soybean Landraces Collected from Zhejiang Province in China” by Yu et al.
1. It will be nice to add ANOVA analysis and broad sense heritability estimation for seed size (100-seed weight), protein content, oil content, and sugar content based on the data from this study.
2. It will be nice to add the figures for the distributions of seed size (100-seed weight), protein content, oil content, and sugar content.
3. It will be nice to add correlation analysis among seed size (100-seed weight), protein content, oil content, and sugar content.
4. It will be nice to add phenotype diversity and differentiation analyses based on the phenotypic data.
5. It will be nice to add principal component analysis (PCA) for the 78 soybean accessions based on (1) phenotypic data and (2) SSR data.
6. It will be nice to perform a population structure analysis among the 78 accessions using SSR data.
7. For the Figure 1 “Clustering of the soybean landraces based on SSR markers”, it seems there are mainly three clusters (sub-populations). Usually, if a cluster or sub-population has less than 5 entities (lines, accessions), the cluster (or sub-population) is merged into a larger one or is recognized as an outlier. Please re-divide the clusters combining PCA and phylogenetic analysis.
8. It is interesting to do a comparison between the phylogenetic analysis based on phenotypic data and the SSR data if authors like to! but it is not necessary.
Comments on the Quality of English Language
Please double check the English grammar and the format based on the requirement of the journal!
Round 2
Reviewer 1 Report
Comments and Suggestions for Authors
Line 56 and 57: no “the” in these two lines because “the” will be used to refer to something/someone were mentioned previously.
L57: yellow-seeded?
L61: accumulate should be exchanged to be “introduce”
Figure 1: there was a blue bar above the plant height histogram, what is it for?
L123-124: will you present data for lodging scores or just an overall observation
Also, rephrase the sentence such as “these 78 soybean landraces were susceptible to the lodging at different degree which potentially results in yield reduction and harvest losses”
L157: cluster, not custer
L164-167: should this part be deleted as it contradict with the previous sentence in Line 163. You stated there are two groups and why there are group VII and VIII
L171-175: What are the purpose of mentioning these landraces? The oil and protein content of these landraces are very typical of soybeans, not particular higher so I don’t understand why these lines were mentioned.
L202: of the landrace lines assessed.
L204: based on
L210-211: The last sentence is against the intention of this work. I think the authors should suggest the use of landraces with superior performance for breeding, not exchanging breeding lines. It is a correct statement but not in a right context.
Suggestions: in the discussion, the points raised were too general. You need to discuss the results of your study more specifically. I suggested to add a paragraph discussing a few soybean lines with superior phenotypic values (oil and protein content) and their agronomic traits which may show the potential/challenge for use for breeding (ie Line ABC has 48% protein content but black seed coat and tall plants which would take longer time to get rid of the seed colour and lodging)
Comments on the Quality of English Language
Please review one more time and check on normal language use such as "the", they are overused in the manuscript
